# Susceptibility of Field Populations of Eggplant Fruit and Shoot Borer (*Leucinodes orbonalis* Guenée) to Cry1Ac, the Protein Expressed in Bt Eggplant (*Solanum melongena* L.) in Bangladesh

**DOI:** 10.3390/insects10070198

**Published:** 2019-07-05

**Authors:** Md. Zulfikar Haider Prodhan, Dattatray K. Shirale, Md. Zaherul Islam, Md. Jahangir Hossain, Vijay Paranjape, Anthony M. Shelton

**Affiliations:** 1Tuber Crops Research Sub Centre, BARI, Bogura 5801, Bangladesh; 2Mahyco Research Center, Maharashtra Hybrid Seed Company Pvt. Ltd. (Mahyco), Post Box No 76, Jalna 431213, India; 3On Farm Research Division, BARI, Pabna 6600, Bangladesh; 4Country Coordinator for Bangladesh, USAID Feed the Future South Asia Eggplant Improvement Partnership, Dhaka 1206, Bangladesh; 5Sathguru Management, Hyderabad 500034, India; 6Department of Entomology, Cornell University AgriTech, Geneva, NY 14456, USA

**Keywords:** Bt, genetic engineering, resistance, eggplant, brinjal

## Abstract

Eggplant (*Solanum melongena* Linn.), or brinjal, was engineered to express an insecticidal protein (Cry1Ac) from *Bacillus thuringiensis* (Bt) and commercialized in Bangladesh on a limited basis in 2014. As part of an insect resistance management strategy, studies were conducted to determine the susceptibility of the targeted insect pest, the eggplant fruit and shoot borer, *Leucinodes orbonalis* (Guenée), to Cry1Ac using a diet-incorporation bioassay method. Eighteen populations of *L. orbonalis* were collected from the main brinjal growing areas in 17 districts of Bangladesh during 2018–2019 and assayed. Larvae from each population were reared to adults and allowed to mate. Eggs from the matings were allowed to hatch, and neonates were used for bioassays. Bioassays were performed with different concentrations of Cry1Ac and an untreated control. Median lethal concentrations (LC_50_) ranged between 0.035 and 0.358 ppm and molt inhibitory concentration (MIC_50_) values ranged from 0.008 to 0.181 ppm. Variation in susceptibility among field populations was 10.22-fold for LC_50_ and 22.63-fold for MIC_50_. These results were compared to values from 73 populations in India. Overall, the results showed similar natural variation and suggest that these Bangladeshi values can be used as benchmarks for resistance monitoring as Bt brinjal becomes more widely adopted in Bangladesh.

## 1. Introduction

Eggplant, or brinjal (*Solanum melongena* Linn.), is the third most important vegetable in terms of production and was grown on 50,956 hectares in Bangladesh during 2016–2017 [1]. The crop is attacked by the eggplant or brinjal fruit and shoot borer (*Leucinodes orbonalis* Guenée) (Lepidoptera: Crambidae), and the percent of fruit infested can reach 90.86% [2]. It has been reported that 98% of Bengali farmers relied solely on insecticide applications to control *L. orbonalis* [3], and farmers spray insecticides several times per week with as many as 84 applications during a 6 to 7 month cropping season [4]). 

As an alternative to intensive use of insecticides, the India-based Maharashtra Hybrid Seed Company Pvt. Ltd. (Mahyco, Dawalwadi, India) inserted the *cry1Ac* gene, under the control of the constitutive 35S CaMV promoter, into brinjal and developed event EE-1 to control feeding damage by *L. orbonalis*. The *cry1Ac* gene has a long history of safe use in the environment [5]. Bt brinjal demonstrated control of *L. orbonalis* in contained greenhouse trials in India [6]. In late 2003, a partnership was formed between Mahyco, Cornell University, the United States Agency for International Development (USAID), and public sector partners in India, Bangladesh, and the Philippines under the Agricultural Biotechnology Support Project II [7]). Mahyco donated the EE-1 event to the Bangladesh Agricultural Research Institute (BARI) and incorporated it into BARI-developed local varieties. BARI conducted confined field trials that demonstrated the Bt lines provided excellent control of *L. orbonalis* compared to their non-Bt isolines [8]. Regulatory assessment of event EE-1 demonstrated its safety for use as human food and animal feed, with no adverse effects to the environment. On October 30, 2013, Bangladesh approved four Bt brinjal varieties for commercial cultivation and Bt brinjal seedlings were distributed to 20 farmers in 2014 [6]. 

A two-year field experiment (2016–2017) conducted by BARI scientists compared the four Bt lines to their isolines, with and without insecticide treatments [9]. Results indicated that Bt varieties had increased fruit production and minimal *L. orbonalis* fruit infestation compared with their respective non-Bt isolines. Fruit infestation for Bt varieties varied from 0 to 2.27% in 2016 and was 0% in 2017, and was not significantly affected by the spray regime in either year. Even in the infested fruit, only small pinprick-like damage was observed on the skin and no live larvae were found. In contrast, fruit infestation in non-Bt lines reached 36.70% in 2016 and 45.51% in 2017, even with weekly spraying, and live larvae were easily found. In this study, an economic analysis revealed that all Bt lines had higher gross returns than their non-Bt isolines and the non-sprayed, non-Bt isolines resulted in negative returns in most cases. Furthermore, this study revealed that statistically similar densities of non-target arthropods, including beneficial arthropods, were generally observed in both Bt and non-Bt varieties, indicating that Bt brinjal could help maintain arthropod biodiversity.

To maintain the economic and environmental benefits of Bt brinjal, it is important that the target insect, *L. orbonalis*, does not rapidly evolve resistance to the Cry1Ac protein. Several deployment tactics to delay resistance have been proposed as insect resistance management (IRM) strategies for Bt crops. These tactics include the use of a high dose of the insecticidal toxin, the number of toxins co-expressed in a single plant, the expression of the toxin in various plant tissues against insect pest(s), planting non-Bt plants as a refuge for Bt-susceptible alleles in the pest population, and monitoring for changes in the susceptibility of the target insect pest to the expressed protein [10]. In the case of Bt brinjal in Bangladesh, a two-gene event is not yet available, but farmers are being trained to plant a 5% structured refuge with non-Bt plants [6]. Several methods have been proposed for monitoring changes in susceptibility. The ‘Holy Grail’ of monitoring techniques may be the development of an appropriate molecular technique that can detect the presence of a resistant allele, but it is unlikely that a single molecular test will be suitable for detecting Cry1Ac-resistance-allele frequency in all insect species or perhaps even in populations within a species [10]. 

Currently, the most practical monitoring methods include evaluating the susceptibility of insects collected in the field against a dose of the toxin in a laboratory diet bioassay, or assessing crop damage or surviving larvae in the field [11]. Dose-bioassay studies consist of incorporating a range of concentrations of the toxin, or a single well-defined diagnostic concentration, into an artificial diet and assessing survival or molting inhibition of the treated insects. The resulting data can be described as the dose that will provide some effect (death or inability to molt) on a proportion of the population (e.g. the LC_50_ value) or as a diagnostic or discriminating dose that will allow only resistant individuals to survive. Such data can then be compared to data from subsequent bioassays to assess changes in susceptibility.

In this study, we report the first set of bioassays on the susceptibility to Cry1Ac of field-collected *L. orbonalis* populations in Bangladesh.

## 2. Materials and Methods

Insect populations: Non-Bt brinjal fruits infested with *L. orbonalis* were collected from 17 different brinjal growing districts in Bangladesh. At least 100 larvae were collected from each of the following sites: Pabna (two locations), Kushtia, Sirajganj, Bogura, Rangpur, Joypurhat, Gaibandha, Rajshahi, Naogaon, Natore, Jessore, Faridpur, Dinajpur, Meherpur, Rajbari, Chaudanga and Sylhet during 2018–2019 (Table 1). These areas represent the main brinjal production regions in Bangladesh. Larvae were reared to adults on a semi-synthetic diet and were allowed to mate using standardized rearing techniques [12] in the laboratory. Eggs collected from matings were allowed to hatch, and the neonate larvae of these field-collected populations (F_1_) were used in bioassays as described below.

Bioassays: Bioassays compared the susceptibility of larvae (F_1_) of the 18 populations and involved exposure of neonate larvae to concentrations of the Cry1Ac protein incorporated into an artificial diet to produce 0–100% mortality. The stock solution (250 ppm) of Cry1Ac was made in a 0.2% agar solution and dilutions were made in deionized water. The source of the Cry1Ac protein used in the bioassays was the commercial formulation MVP ll^R^ (Mycogen Crop., USA), which contains 19.7% (by weight) Cry1Ac. Various concentrations of Cry1Ac were later mixed into the semi-synthetic diet and approximately 750 uL of Cry1Ac mixed diet was poured into each well of a 128-well bioassay tray. Using a fine brush, one neonate was placed in each well. For each population, seven concentrations and one control were included with 16 insects per treatment, and treatments were replicated three times, resulting in at least 384 insects tested per location. Bioassay trays were kept in a growth chamber at 25 + 1°C and 55–65% relative humidity. 

Observations: Observations were recorded on mortality and instar stage of the surviving larvae after seven days. Larvae that did not move when disturbed were considered to be dead. Larvae that failed to molt to the next stage were considered as being inhibited. 

Statistical analysis: The mortality data was subjected to probit analysis using POLO-PC [13] to estimate the lethal concentration (LC) and molting inhibitory concentration (MIC). The concentration of Cry1Ac protein that will kill 50% of the test population over a given period of time is termed the LC_50_. The concentration that will inhibit 50% of neonates to become second instars is called the molting inhibitory concentration (MIC_50_). The range of variation in both these values was assessed by calculating the number of ‘folds’ between the lowest and highest values.

Fold variability: Fold variability was calculated using the following formula:
(1)$$Fold\;variability=\;\frac{Highest\frac{LC}{MIC}values\;exhibited\;by\;a\;population}{Lowest\frac{LC}{MIC}\;values\;exhibited\;by\;a\;population}.$$

## 3. Results

The susceptibility of *L. orbonalis* populations exposed to various concentrations of the Cry1Ac protein are described below (Table 2). The LC_50_ values ranged from 0.035 to 0.358 ppm of diet and there were significant differences between populations. The most susceptible was the Mohadebpur-Naogaon population (LC_50_ value 0.035), and the least susceptible was the Sadar-Rangpur population (0.358). There was a 10.2-fold variation in LC_50_ values of the 18 populations tested for Cry1Ac susceptibility. Similarly, the LC_95_ values of the populations ranged from 0.647 to 6.936 ppm, with a 10.7-fold variation among the populations, and there were significant differences between populations.

The MIC_50_ values for 18 locations ranged from 0.008 (Atghoria- Pabna; Gangni-Meherpur) to 0.181ppm (Sadar-Rangpur), a range of 22.6-fold, and there were significant differences between populations (Table 3). The MIC_95_ values displayed a 48.5-fold inter-population variation. 

## 4. Discussion

These data represent what we consider to be the natural variation in the susceptibility of field populations of *L. orbonalis* to Cry1Ac in Bangladesh. Even though Bt brinjal has been grown in limited amounts in Bangladesh beginning in 2014 and has been increasingly adopted by farmers, the overall adoption of Bt brinjal represents a small percentage (<7%) of the total brinjal area in 2018. Furthermore, it is estimated that there are >100 local varieties grown and only four of them have been engineered to express Cry1Ac. Thus, from a landscape perspective, these 18 populations should be considered to represent the natural range of susceptibility.

Our results on susceptibility to the Cry1Ac protein can be compared to previous studies conducted in India over several years using the same protocols. For 29 *L. orbonalis* populations tested in 2004–2005, the LC_50_ values ranged from 0.008 to 0.095 ppm, an 11.9-fold range [14]. In 2007–2008, 10 populations were tested and LC_50_ values ranged from 0.019 to 0.040 ppm, a 2.1-fold range [15]. In 2008–2009, another 10 populations were tested and LC_50_ values ranged from 0.019 to 0.037 ppm, a 1.9-fold range [16]). Using similar diet bioassays, populations of *L. orbonalis* collected in 2009 and 2010 from 14 districts in South India had LC_50_ values that ranged from 0.020 to 0.042 ppm, a 2.1-fold difference [17]. In another study from North Karanataka, India, in 2009 and 2010, 10 populations of *L. orbonalis* were tested and LC_50_ values ranged from 0.026 to 0.104 ppm, a 4.0-fold variation [18]. Collectively, these five studies include 73 populations collected from India that displayed LC_50_ values from 0.008 to 0.104 ppm, a 13.5-fold range. With such a range, it is not surprising that there were significant differences in LC_50_ values for some populations. 

The LC_50_ values of the 18 Bangladeshi populations varied from 0.035 to 0.358 ppm, a 10.2-fold variation. However, five of the 18 values obtained in Bangladesh fall outside the highest value obtained in India (0.104). The highest LC_50_ value obtained in Bangladesh (0.358) was 3.3-fold higher than the highest value in India. A possible explanation for this is that the Bangladeshi and Indian studies were performed in different labs. The Indian studies were mostly conducted by Mahyco personnel. Although Mahyco personnel trained the Bangladeshi personnel, slight differences in methodology may have occurred and resulted in differences in outcomes. However, within either bioassay program, one would expect the values to be consistently higher or lower than in the other program, and they were. Over the 73 populations tested in India, the range in values was 13.5-fold, while the range in values for the 18 Bangladeshi populations was 10.2-fold. Such close values in ranges suggest similar natural variation in susceptibility. Another possibility is that there are inherent differences in the populations due to the history of exposure to Cry1Ac. However, in both countries, Cry1Ac plants or sprays containing Cry1Ac were not widely used before these collections were made [6], so this possibility seems unlikely.

A study conducted in the Philippines [19]) used a diet-overlay bioassay to evaluate populations of *L. orbonalis* collected from nine eggplant production areas in the Philippines between 2012 and 2013. Although their data cannot be directly compared to the Indian or Bangladeshi studies because of different methods, they observed a 4.6-fold variation in LC_50_ values. Their observed variation fits within the ca. 10-fold variation seen in India and Bangladesh and should be considered natural variation in Philippine populations. 

Based on the data obtained from India and the Philippines and the results obtained from Bangladesh, we will consider these Bangladesh data to represent natural variation in susceptibility levels to Cry1Ac and to which future studies can be compared. Besides these present studies, additional efforts will be undertaken to develop other monitoring approaches, including developing a discriminating or diagnostic dose. Such a dose may be a more efficient method for monitoring changes in susceptibility and has the added benefit of quantifying the percentage of individuals that are resistant. From a practical standpoint, it will be most efficient to combine insect assays, using either a dose-mortality range or a discriminating dose, with field scouting for crop damage and surviving larvae [11].

Data on susceptibility should be considered as part of an overall stewardship program wherever Bt eggplant is grown. Such data were first collected in India in 2004–2005 [14] where it was anticipated that Bt eggplant would first be released. Beginning in 2004, Bt hybrids were tested in several states in India and performed well, leading Krishna and Qaim [20] to write, “several Bt hybrids have been tested in the field and are likely to be commercialized in the near future.” After extensive field trials and safety evaluations by Indian regulatory bodies, Bt eggplant was ready to be commercialized in India. However, this was disrupted by anti-biotech activities that resulted in the Indian Minister of the Environment and Forests, the last “gatekeeper” before Bt eggplant would be commercialized, to impose a moratorium on Bt eggplant on 9 February 2010, which remains today [21]. Thus, Bangladesh is presently the only country in which Bt eggplant is allowed to be grown.

## 5. Conclusions

Bangladesh has introduced insect-resistant, genetically engineered brinjal that produces the insecticidal crystal protein Cry1Ac, i.e., Bt brinjal. Studies have shown that it provides virtually complete control of *L. orbonalis* [9], dramatically reduces insecticide use, and provides a six-fold increase in net return [22]. Studies on Cry1Ac eggplant in Bangladesh [9] and the Philippines [23] have also demonstrated that it does not disrupt non-target arthropods. Such findings complement other studies that have demonstrated the compatibility of Bt crops and biological control in integrated pest management (IPM) programs [24].

To preserve this valuable technology, it is important that *L. orbonalis* does not become resistant to Cry1Ac. An important step is to monitor natural variation in susceptibility to Cry1Ac for future comparisons when Bt brinjal is more widely adopted. Data presented in this study of 18 populations from Bangladesh can serve that function. The Bangladeshi data compare favorably to data collected from India, and therefore provide confidence in their use.

## Figures and Tables

**Table 1 insects-10-00198-t001:** Details of *Leucinodes orbonalis* populations collected from different brinjal growing districts of Bangladesh in 2018–2019.

Population (District)	Co-ordinates	Sampling Date	Larvae Collected (Number)
Sadar (Pabna)	89°24′49.6456″ E 24°1.58′4158″ N	21 October 2018	100
Atgoria (Pabna)	89°13′13.18119″ E 23°55′41.45033″ N	28 March 2018	130
Kumarkhali (Kushtia)	89°13′35.69283″ E 23°55′41.45033″ N	21 March 2018	110
Enayetpur (Sirajganj)	89°40′28.41276″ E 24°13′46.02393″ N	28 April 2018	124
Sadar (Bogura)	89°25′11.1″ E 24°50′39.33096″ N	24 April 2018	120
Sadar (Rangpur)	89°10′350″ E 25°41′054″ N	08 July 2018	115
Panchbibi (Joypurhat)	89°1′55.10057″ E 25°10′27.04416″ N	03 July 2018	135
Saghata (Gaibandha)	89°35′11″ E 25°5′58″ N	22 May 2018	145
Puthia (Rajshahi)	88°46′71″ E 24°23′024″ N	10 August 2018	120
Mohadebpur (Noagaon)	88°45′34.71064″ E 24°55′49.14894″ N	04 August 2018	105
Lalpur (Natore)	89° 0′38.50574″ E 24° 13′24.87066″ N	24 July 2018	140
Sadar (Jessore)	89°09′26.2″ E 23°13′38.6″ N	15 September 2018	125
Sadar (Faridpur)	89°50′10.61338″ E 23°35′20.70488″ N	14 October 2018	120
Sadar (Dinajpur)	88° 69′ 57.12″ E 25°72′91.53″ N	03 November 2018	120
Gangni (Meherpur)	88° 52′ 80.14279″ E 23°53′40.96221″ N	9 January 2019	115
Sadar (Rajbari)	89° 39′52.5452″ E 23°41′15.6328″ N	20 January 2019	100
Sadar (Chuadanga)	88° 49′ 27.75138″ E 23°37′45.89172″ N	22 January 2019	105
Sadar (Syhlet)	91° 51′12.39″ E 24°47′19.64″ N	23 April 2019	110

**Table 2 insects-10-00198-t002:** Probit analysis of mortality (LC values) of *Leucinodes orbonalis* neonates to diet-incorporated Cry1Ac protein (ppm). Populations were collected in Bangladesh in 2018–2019.

Population (District)	n	χ (df) ^a^	Slope (SE)	LC_50_ (95% CI) ^b^	LC_95_ (95% CI) ^b^
Sadar (Pabna)	384	4.42 (5)	1.12 ± 0.14	0.061 (0.034-0.096)	1.798 (0.957-4.729)
Atgoria (Pabna)	384	2.87 (5)	1.05 ± 0.14	0.055 (0.026-0.095)	1.969 (1.008-5.586)
Kumarkhali (Kushtia)	384	4.61(5)	1.16 ± 0.23	0.118 (0.039-0.218)	3.059 (1.519-12.069)
Enayetpur (Sirajganj)	384	7.36 (5) *	1.50 ± 0.29	0.060 (0.003-0.13)	1.020 (0.460-17.660)
Sadar (Bogura)	384	5.06 (5) *	1.32 ± 0.27	0.130 (0.020-0.260)	2.280 (1.050-17.690)
Sadar (Rangpur)	384	8.85 (5) *	1.28 ± 0.17	0.358 (0.037-1.024)	6.936 (2.104-420.71)
Panchbibi (Joypurhat)	384	8.51 (5) *	1.82 ± 0.39	0.293 (0.078-0.481)	2.346 (1.315-13.432)
Saghata (Gaibandha)	384	10.51 (5) *	1.28 ± 0.17	0.191 (0.121-0.358)	2.502 (1.201-9.587)
Puthia (Rajshahi)	384	17.49 (5) *	1.02 ± 0.15	0.056 (0.004-0.159)	2.313 (0.706-73.995)
Mohadebpur (Noagaon)	384	5.64 (5) *	1.30 ± 0.17	0.035 (0.016-0.062)	0.647 (0.322-2.192)
Lalpur (Natore)	384	10.36 (5)	1.32 ± 0.18	0.069 (0.015-0.154)	1.213 (0.481-10.975)
Sadar (Jessore)	384	16.04 (5) *	0.96 ± 0.12	0.042 (0.003-0.139)	2.291 (0.542-26.74)
Sadar (Faridpur)	384	17.24(5) *	1.07 ± 0.14	0.061 (0.005-0.184)	2.099 (0.562-19.350)
Sadar (Dinajpur)	384	12.34 (5) *	1.21 ± 0.16	0.078 (0.010-0.200)	2.752 (0.566-59.78)
Gangni (Meherpur)	384	19.82 (5) *	1.88 ± 0.11	0.040 (0.002-0.146)	2.862 (0.556-12.591)
Sadar (Rajbari)	384	21.69 (5) *	1.02 ± 0.14	0.048 (0.011-0.150)	3.032 (0.606-446.85)
Sadar (Chuadanga)	384	16.14 (5) *	1.02 ± 0.14	0.053 (0.003-0.171)	2.150 (0.557-252.46)
Sadar (Syhlet)	384	18.22(5) *	0.93 ± 0.12	0.057 (0.003-0.199)	3.214 (0.694-10.523)

^a^ Chi-square goodness of fit as determined using POLO-PC. A value followed by * is significantly different (*P* < 0.05). ^b^ ppm Cry1Ac in diet with 95% confidence interval (CI) at the 50% and 95% levels of probit mortality.

**Table 3 insects-10-00198-t003:** Probit analysis of molting inhibitory concentrations (MIC values) of *Leucinodes orbonalis* neonates to diet-incorporated Cry1Ac protein (ppm). Populations were collected in Bangladesh in 2018–2019.

Population (District)	n	χ (df) ^a^	Slope (SE)	MIC_50_ (95% CI) ^b^	MIC_95_ (95% CI) ^b^
Sadar (Pabna)	432	2.38 (5)	1.38 ± 0.27	0.057 (0.019-0.104)	0.878 (0.488-2.514)
Atgoria (Pabna)	384	8.12 (5) *	2.33 ± 0.34	0.008 (0.004-0.012)	0.039 (0.002-0.146)
Kumarkhali (Kushtia)	384	7.79 (5) *	1.98 ± 0.29	0.010 (0.004-0.218)	0.063 (0.033-0.258)
Enayetpur (Sirajganj)	384	0.35 (5)	1.75 ± 0.25	0.009 (0.006-0.013)	0.079 (0.050-0.162)
Sadar (Bogura)	384	1.24 (5)	2.12 ± 0.41	0.016 (0.008-0.024)	0.098 (0.064-0.208)
Sadar (Rangpur)	432	9.14 (5) *	1.22 ± 0.21	0.181 (0.016-0.446)	1.891 (1.414-9.120)
Panchbibi (Joypurhat)	432	7.72 (5) *	1.83 ± 0.35	0.180 (0.035-0.316)	1.436 (0.749-13.761)
Saghata (Gaibandha)	432	14.42 (5) *	1.74 ± 0.57	0.145 (0.041-2.159)	1.276 (0.412-15.231)
Puthia (Rajshahi)	384	4.94 (5)	1.61 ± 0.27	0.027 (0.013-0.043)	0.878 (0.488-2.514)
Mohadebpur (Noagaon)	384	3.56 (5)	1.91 ± 0.28	0.019 (0.012-0.026)	0.137 (0.091-0.263)
Lalpur (Natore)	384	1.58 (5)	2.07 ± 0.30	0.028 (0.015-0.038)	0.171 (0.115-0.319)
Sadar (Jessore)	384	1.84 (5)	1.70 ± 0.23	0.011 (0.007-0.015)	0.097 (0.062-0.194)
Sadar (Faridpur)	384	3.08 (5)	1.94 ± 0.26	0.017 (0.011-0.023)	0.119 (0.079-0.220)
Sadar (Dinajpur)	384	1.31 (5)	1.99 ± 0.27	0.016 (0.010-0.022)	0.149 (0.073-0.299)
Gangni (Meherpur)	384	0.89 (5)	1.49 ± 0.21	0.008 (0.005-0.012)	0.106 (0.065-0.227)
Sadar (Rajbari)	384	4.61(5)	1.43 ± 0.17	0.012 (0.008-0.017)	0.166 (0.102-0.340.
Sadar (Chuadanga)	384	7.70 (5) *	1.44 ± 0.19	0.011 (0.004-0.021)	0.157 (0.074-0.730)
Sadar (Syhlet)	384	1.19 (5)	1.47 ± 0.17	0.016 (0.010-0.022)	0.206 (0.128-0.415)

^a^ Chi-square goodness of fit as determined using POLO-PC. A value followed by * is significantly different (*P* < 0.05). ^b^ ppm Cry1Ac in diet with 95% confidence interval (CI) at the 50% and 95% levels of probit mortality.

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
