# Peer review of "Susceptibility of Field Populations of Eggplant Fruit and Shoot Borer (Leucinodes orbonalis Guenée) to Cry1Ac, the Protein Expressed in Bt Eggplant (Solanum melongena L.) in Bangladesh"

_insects, 2019, doi:10.3390/insects10070198_

Round 1
Reviewer 1 Report
This is a relatively simple study that will have considerable value in the literature. I do have a couple of suggestions, but the paper is largely acceptable in the current format.
Comments:
Line 183-188, omit: “A possible explanation for this is that the Bangladesh and Indian studies were performed in different labs. TheIndian studieswere mostly conducted by Mahyco personnel. Although Mahyco personnel trained the Bangladeshi personnel, slight differences in methodology may have occurred and resulted in differences in outcomes. However, within either bioassay program one would expect the values to be consistently higher or lower than in the other program, and they were”. This claim is made again in on lines 190-191, and once is enough.
The paper has an unusually heavy reliance on non-peer-reviewed citations (presentations, annual reports, news reports, etc.). Given the paucity of peer-reviewed data available from Bangladesh there may be few alternatives, but the citations lacking peer review should be minimized to the degree possible. However, this problem highlights the need to produce peer-reviewed manuscripts like this one to document baseline data on Cry1 Ac resistance as well as potential costs and benefits of the introduction of GM crops.
Author Response
Reviewer #1.
We appreciate your remark that this paper will “have considerable value in the literature” and that it is “largely acceptable in its current format.”
As far as our “heavy reliance on non-peer reviewed citations,” we appreciate your recognition that this highlights the need to produce peer-reviewed manuscripts, like this one, to document baseline data….”
We have eliminated the repeat about the discussion on different labs. We have also included more details on the bioassay methods and done additional editing to clarify points.
Reviewer 2 Report
The manuscript reports the results of laboratory bioassays investigating the susceptibility of Leucinodes orbonalis larvae to a Cry1Ac based product. This study is very preliminary and provides information from the laboratory (artificial diet bioassays), while it refers (in the introduction and discussion) to field conditions (Bt-plants).
The methods are not clearly presented so that it is difficult to evaluate the results reliability (see below).
Such data, if available, needs to be provided.
The manuscript needs a significant revision (including English language) before being considered for publication. I recommend rejection of the manuscript in the present form.
INTRODUCTION
Introduction is dispersive. Information and details on the Bt and non-Bt plants should be reduced, while this section should focus on the aspects that are directly related to the object of the study. The introduction should clearly state what are the original features of the research.
MATERIALS AND METHODS
L95-100: Develop this section. Explain here that bioassays compared susceptibility of larvae (F1) of different origin. How many larvae form each location? Without this number it is not possible to evaluate the result reliability. A good number could be for example 10 larvae x 4-5 replicates for each concentration tested. You need at least 5-7 concentrations and the whole experiments should be repeated three times to calculate a reliable LC50.
L114-115: it is not clear how many larvae (one per well?) for each concentration. Then, state here how many and which concentrations were assayed.
RESULTS and DISCUSSION
The reliability of results has to be weighed on the basis of the methods employed. They have to be reconsidered after providing information on replicates and concentrations assayed.
Author Response
We appreciate the time you spent reviewing the manuscript and your suggestions for improvements. We make the following replies.
Materials and Methods. We have added details on the number of doses (7) and the untreated control (1), and the number of insects used per location (348). We already had included the number of larvae per well (1) but clarified it.
Introduction. The comment was made that this section is “dispersive” and that it “should focus on the aspects that are directly related to the object of the study.” The “dispersive” comment suggests the contents of the section are scattered and don’t follow a logical flow of ideas. We disagree and think that the background on brinjal production, and Bt brinjal in particular, set the stage for the “focus” of the paper, which is to document baseline susceptibility of the insect to Cry1Ac. Setting the background for the “focus” of the study only required 5 paragraphs and we prefer to keep this concise background into the paper.
Results and Discussion. We trust that providing more details on the methods we used will provide the reader with confidence in the results.
Overall. We have edited the manuscript for grammar and content.
Reviewer 3 Report
The manuscript describes the survey regarding susceptibility of field populations of Leucinodes orbonalis collected from different eggplant growing districts of Bangladeshi to Cry1Ac.
According to the authors description in Materials and Methods lines 95 to 102, the larvae used for bioassay were F1 collected from the non-Bt brinjal fruits. Therefore, all the larvae from 11 districts are expected to be Cry1Ac sensitive. I wonder why authors did not test the larvae from the Bt brinjal fruits, which should be their main concern. Also, there is no negative control in the data presented (for example, Cry3A).
In Table 1, combine the columns of Population and District just like Tables 2 and 3. It is confusing.
Author Response
There were no larvae that could be found in Bt brinjal fruits or shoots because they were killed by the Cry1Ac protein. Larvae were only collected from non-Bt fruit. In theory and practice, insects collected from the non-Bt plants could be susceptible or resistant and would represent the status or frequency of resistance in the population in that field. Collecting insects this way is common practice.
Cry3A only works against coleopteran insects so would not provide any additional information about susceptibility to Cry1Ac (see review by Schnepf et al. 1998, https://mmbr.asm.org/content/62/3/775). Using 7 doses, plus a water control, provided excellent results using probit analysis.
Thank you for your suggestion about combining the columns for population and district in Table 1.
Round 2
Reviewer 2 Report
The manuscript has been improved from previous version and the methodological concerns have been clarified. I recommend publication of the present version.
Author Response
Thank you.
Reviewer 3 Report
N/A